# The Spectrum of Clinical and Serological Features of COVID-19 in Urban Hemodialysis Patients

**DOI:** 10.3390/jcm9072264

**Published:** 2020-07-16

**Authors:** Teresa Stock da Cunha, Elena Gomá-Garcés, Alejandro Avello, Mónica Pereira-García, Sebastian Mas-Fontao, Alberto Ortiz, Emilio González-Parra

**Affiliations:** 1Nephrology and Hypertension Department, Fundación Jimenez Díaz, 28040 Madrid, Spain; teresa.stock@quironsalud.es (T.S.d.C.); elena.goma@quironsalud.es (E.G.-G.); alejandro.avello@quironsalud.es (A.A.); aortiz@fjd.es (A.O.); 2Iñígo Álvarez de Toledo Renal Foundation (FRIAT), 28040 Madrid, Spain; mpereira@friat.es; 3Renal, vascular and diabetes research laboratory, IIS-Fundación Jimenez Díaz/UAM, 28040 Madrid, Spain; Smas@fjd.es; 4Spanish Biomedical Research Centre in Diabetes and Associated Metabolic Disorders (CIBERDEM), 28040 Madrid, Spain; 5Spanish Renal Research Network (REDinREN), 28040 Madrid, Spain; 6UAM Medical School, Universidad Autonoma de Madrid (UAM), 28040 Madrid, Spain

**Keywords:** COVID-19, hemodialysis, chronic kidney disease, antibodies, asymptomatic, mortality

## Abstract

Introduction: The inherent immunosuppression of uremia increases the susceptibility of hemodialysis patients to infection. There is still limited evidence on hemodialysis patients and COVID-19. The clinical and analytical spectrum and treatment responses and mortality are poorly characterized. Material and Methods: Clinical and analytical features, chest X-ray, polymerase chain reaction (PCR) and antibodies for SARS-CoV-2, treatment and outcomes were analyzed in 48 patients diagnosed with COVID-19 during March and April 2020 in two coordinated Spanish hemodialysis units. Results: In 200 haemodialysis patients, COVID-19 was diagnosed in 48, of whom 22 were PCR positive, eight PCR negative but seroconverted and two were diagnosed on typical clinical grounds. Despite a mean age of 72.6 years, the overall mortality rate was 5/48 (10%). Among the PCR positive patients, 21 (55%) required admission and five (13%) died. PCR positive patients were more often symptomatic and hospitalized and had higher troponin I levels than PCR negative patients, but did not differ in lymphocyte counts, D-dimer or interleukin-6 (IL-6) levels. Among PCR negative COVID-19 patients, three out of 10 (30%) required admission, and none died. The most frequent symptom among the 48 patients was fever (31%), followed by asymptomatic patients (23%). A low number of lymphocytes was the only parameter significantly different between hospitalized and ambulatory COVID-19 patients, independently of PCR status. Conclusions: COVID-19 hemodialysis patients are frequently asymptomatic, and mortality may be lower than previously reported. Diagnosis may be retrospective, based on seroconversion, as PCR may be negative. This information should guide preventive and patient isolation strategies.

## 1. Introduction

In February 2020, the SARS-CoV-2 coronavirus epidemic that began in late December in Wuhan, China, reached Madrid. Hemodialysis patients have higher mortality from any cause than the general population. The leading cause of death is cardiovascular, followed by infections [1,2]. Lung infections are the most common cause of infectious death [3] and sepsis mortality is 50 times higher than in the general population [4]. Altered innate and adaptive immunity in uremia increases the risk and severity of infections [5]. The increased sensitivity to infection coexists with a state of systemic inflammation [6,7]. Infections and death from cardiovascular causes are interrelated, so that the risk of death from cardiovascular causes increases markedly after respiratory infections, in both the general population and dialysis patients [8,9]. Specifically, mortality from viral causes is high in hemodialysis, as the seasonal influenza virus can be complicated by pneumonia, bronchitis and myocarditis, among others [10]. As the influenza virus is highly contagious, annual vaccination is recommended, despite the suboptimal immune response of dialysis patients [11]. Six months after vaccination, only one-third of hemodialysis patients maintain immunity [12]. These features of dialysis patients suggest that coronavirus disease 2019 (COVID-19) may be more severe in hemodialysis patients. COVID-19 is a highly contagious, slow course and potentially lethal disease that is more severe in patients with comorbidities such as diabetes, cardiovascular disease and chronic kidney disease [13].

COVID-19 is characterized by high virus shedding from the upper respiratory tract even among asymptomatic patients, which differentiates it from influenza [14,15,16]. Symptom-based detection alone could not detect a high proportion of infectious individuals, failing to control transmission in closed communities. Thus, it was recently recommended to perform routine polymerase chain reaction (PCR) on contacts and newcomers, in addition to the symptomatic detection of new residents and the periodic reassessment of the study population and staff, following a strategy similar to that successfully containing the epidemic in South Korea, Singapore and Germany.

Information on the clinical course of COVID-19 in hemodialysis patients is scarce. In one hemodialysis unit in China, there were 37 (16%) cases of COVID-19 among 230 patients and seven out of 37 (19%) COVID-19 patients died [17]. In addition to lung lesions identified on a computed tomography (CT) scan, the disease was accompanied by leukopenia and increased inflammatory cytokines. This study alerted us to the high susceptibility of hemodialysis patients to and the high mortality of COVID-19.

We now present the experience in a Spanish Dialysis Unit, collected two months after the first COVID-19 case, and analyze the clinical and analytical characteristics, the immune response to the virus, treatment and outcomes.

## 2. Materials and Methods

This was a retrospective observational study of COVID-19 in chronic hemodialysis patients from two coordinated dialysis units: the Hemodialysis Unit of the Fundación Jiménez Diaz Hospital (UHFJD), Madrid, Spain, and its associated center Fundación Renal Centro Santa Engracia (FRCSE). UHFJD is a hospital-based hemodialysis unit that generally cares for patients with more comorbidity, while FRCSE is a satellite limited care center (providing only hemodialysis outside a hospital setting) with its own management autonomy, which generally cares for patients with a lower comorbidity index. The study was approved by the IIS-Fundación Jiménez Díaz Ethics Committee (PIOH036-20_FJD) and was performed in accordance with the Declaration of Helsinki and the European Union Clinical Trial Directive. Patients were enrolled after providing written informed consent.

In March 2020, UHFJD had 58 patients in a chronic hemodialysis program and FRCSE had 142 additional patients (200 patients in total, aged over 18 years) (Figure 1). Both centers cooperate to care for chronic hemodialysis patients: stable patients are transferred from UHFJD to FRCSE and, if they become unstable, they are bought back to the hospital-based UHFJD. The study was conducted on patients suspected of having an infection, i.e., who presented clinical symptoms compatible with COVID-19 or asymptomatic patients in close contact with individuals with suspected or confirmed active infection. Patients were screened for symptoms daily. Patients from FRCSE with a diagnosis of COVID-19 were transferred to UHFJD and this study was performed among UHFJD patients, which include some transferred from FRCSE, as detailed below.

### 2.1. COVID-19 Diagnosis

Between March and April 2020, the FJD unit had a total of 66 patients, who were all tested for SARS-CoV-2 by PCR. Of these, 27 patients were transferred from FRCSE to UHFJD, as they needed hospital admission (17/27) and/or were positive for COVID-19 as outpatients (10/27). The remaining 39 patients from HUFJD included patients with symptoms (34/39) compatible with COVID-19 or close contact with symptomatic individuals, as well as 5/39 UHFJD patients who were tested, because they had shared the dialysis unit with COVID-19 patients.

The PCR test for SARS-CoV-2 consists of a nucleic acid extraction from nasopharyngeal swab samples and subsequent amplification by real-time RT-PCR using ORF1ab and N genes as targets. Serological (IgM/IgG) tests were also performed using ELISA (VIRCELL, Granada, Spain). Sensitivity: IgG: 100% (19 days after PCR positive), IgM: 82–88% (7 days after PCR positive)/specificity IgG: 98%, IgM: 98.8%.

### 2.2. Additional Tests

A COVID-19-related analytical panel included hemogram, lymphocyte profile, ferritin and interleukin-6 (IL-6) as markers of inflammation, D-dimer as a marker of vascular damage, and Troponin I as a marker of myocardial damage. Those with high suspicion of COVID-19 were subjected to a chest X-ray.

### 2.3. Prevention and Treatment

Droplet respiratory protection measures with a surgical mask for both patients and staff, hand washing and other isolation measures were initiated on February 24, 2020, two weeks before the first COVID-19 patient was diagnosed.

Admission criteria were based on the presence of pneumonia, oxygen saturation below 94% (or hypoxemia evidenced in arterial blood) and/or significant deterioration in general condition. Initial treatment for all admitted patients and outpatients was according to the weekly updated protocol of the FJD. Treatment consisted of hydroxychloroquine (200 mg/12 h for 5 days) and antibiotics (doxycycline 100 mg/12 h for 5 days or levofloxacin 250 mg/48 h for 5 days). In case of acute respiratory insufficiency and inflammatory signs, such as marked elevation of ferritin and C-reactive protein, glucocorticoids were added (250 mg/day Methylprednisolone for 3 days, followed by oral prednisone 40 mg every 12 h for 3–4 days). Prophylactic tinzaparin (3500 IU/day) was prescribed for those with analytical data of procoagulant status, i.e., D-dimer elevation or with markers of acute myocardial damage. For bilateral pneumonia and/or oxygen saturation below 94%, Lopinavir-Ritonavir (Kaletra 400 mg/12 h for 5 days) was added and finally, in the most severe cases, Tocilizumab 400 mg (maximum 2 doses in 48 h) was added, if the oxygen requirement was between 10–15 L/min. Oxygen therapy was initiated when oxygen fell below 93%. The most severe cases were admitted to the ICU. Hemodialysis was performed in an independent isolation room for PCR-positive cases.

### 2.4. Statistical Data Analysis

Quantitative variables were described using the mean and standard deviation or the median and interquartile range. Qualitative variables were described by means of absolute and relative frequencies. Groups were compared with the Student t-test or the Mann-Whitney test, depending on the normality of the data (Shapiro-Wilk test) for quantitative variables, and the Chi-square test for qualitative variables, using R version 4.0 (R Foundation for Statistical Computing) [18].

## 3. Results

### 3.1. COVID-19 Diagnosis

Of 200 patients on hemodialysis distributed between two associated centers, 66 patients were tested for SARS-CoV-2 by PCR between March and April 2020. In addition, anti-SARS-CoV-2 antibodies were studied in 57/66 (86.4%) patients.

In total, 48/200 (24%) patients were diagnosed as COVID-19: 38/48 (79%) based on positive PCR; eight out of 48 (17%) patients with negative PCR were diagnosed based on seroconversion and two (4%) had a clinical diagnosis (Figure 1). The clinical diagnosis despite negative PCR and anti-SARS-CoV-2 antibodies was based on clinical features: they required admission for typical COVID-19 clinical, radiological (unilateral or bilateral pneumonia) and analytical features (very low lymphocyte counts, higher D-dimer values than other hemodialysis non-affected patients and high IL-6 levels) (Appendix A). The mean age of COVID-19 patients was 72.6 years, and 16 (33%) were females (Table 1).

Anti-SARS-CoV-2 antibodies were studied in 29/38 (88%) of PCR-positive patients and in all PCR-negative patients. Among PCR positive patients, 17 (45%) were IgM positive and 17 were IgG positive. Five (13%) PCR positive patients remained with negative serology, a mean of 13.2 ± 5.2 days after PCR positivity. Among PCR-patients, eight out of 28 (28.6%) became positive for IgM and four (11%) were additionally positive for IgG.

### 3.2. Clinical Features of Patients with Positive and Negative PCR

The overall incidence of PCR positive COVID-19 was similar in both centers: 27/142 (19%) in FRCSE, and 11/58 (19%) in UHFJD. FRCSE PCR positive patients were moved to UHFJD. A majority (68.4%) were men and the mean age was 73.4 ± 11.9 years (Table 1, Appendix A). Of PCR positive patients, 21/38 (55.3%) required hospital admission and five out of 38 died (13%). One patient was admitted to the intensive care unit (ICU). Disease was less severe in PCR negative patients, who had a lower need for admission (three admissions: 30%, *p* = 0.42 and no deaths were recorded. Moreover, COVID-19 was more symptomatic in PCR positive patients: 87% had symptoms, as opposed to 40% of PCR negative patients (*p* = 0.001) (Appendix A). The most frequent symptom in both groups of patients was fever found in 36% of PCR positive and 30% of PCR negative patients). Additional presenting symptoms were general malaise, cough, diarrhea, dyspnea, arrhythmia, chest pain and nausea.

PCR positive and negative patients did not display differences in oxygen saturation or in several analytical parameters associated with COVID-19, including low lymphocyte counts, high D-dimer levels or high IL-6 levels. Low lymphocyte counts in patients diagnosed with COVID-19 were not present months earlier. However, serum troponin I levels were higher in PCR positive patients: 0.050 (0.020–0.098) vs. 0.012 (0.001–0.040); *p* = 0.033 (Table 1). Among patients who had chest X-ray, normal chest X-ray, unilateral and bilateral pneumonia was observed in both patients with positive and negative PCR.

### 3.3. Need for Hospitalization

The characteristics of hospitalized and non-hospitalized PCR positive patients are shown in Table 2 and Appendix A and for PCR negative patients in Table 3 and Appendix A. Among PCR positive patients, there was a non-significant numerical towards higher X-ray severity and lower oxygen saturation. However, only lymphocyte counts were significantly lower in hospitalized patients: non-hospitalized 950 (600–1250) vs hospitalized 600 (300–700), *p* = 0.007. This was also the case for PCR negative patients—non-hospitalized: 1000 (900–1350) vs hospitalized: 400 (300–700); *p* = 0.04. A non-significant numerical towards higher serum IL-6 levels was also observed for both hospitalized PCR positive and hospitalized PCR negative patients.

### 3.4. Outcomes

At the end of follow-up, 18/38 (47%) patients with positive PCR have become negative, are asymptomatic for fever, cough and/or dyspnea and are integrated in the general dialysis room; 11/38 (28.9%) are clinically stable, although they persist with positive PCR and are being dialyzed in the COVID-19 isolation unit and four out of 48 (10.5%) were still hospitalized (three in a general ward and one in the ICU).

Overall mortality between March and April 2020 was five out of 48 (10.4%) patients, but it was 30% higher in PCR positive patients: five out of 38 (13.2%), while no PCR negative patient died. The mean age of the deceased was 79 ± 4 years. Interestingly, the highest mortality (two out of six; 33.3%) was found among patients on hemodialysis for less than 3 months. Both patients were immunosuppressed for underlying diseases (myeloma and vasculitis, respectively). By contrast, the mortality among chronic (>3 months) hemodialysis patients was three out of 42 (7.1%). These other three positive PCR patients died from acute myocardial infarction (one) and from respiratory failure (two).

## 4. Discussion

The main findings of this study are that, among hemodialysis patients, there is a high incidence of oligosymptomatic patients, as well as of infected patients with negative PCR results. The disease spectrum is further expanded by patients who are undistinguishable from a clinical, radiological and analytical point of view from bona fide COVID-19 patients, yet are PCR and anti-SARS-CoV-2 antibody negative. In this regard, seroconversion was not observed in all hemodialysis COVID-19 patients. A strategy of early diagnosis and treatment was associated with mortality in the range described for the general population of similar age [18], and lower than for some European hemodialysis series, and in the range found in some Chinese centers caring for younger hemodialysis patients (Appendix A) [17,19,20,21,22].

In the FJD unit, we had a policy of early PCR in patients with suggestive symptoms or having cohabitated with infected individuals. In one month, most patients had been tested, and in two months, the whole unit was tested. The asymptomatic positive patients had had contact with symptomatic family members. Current recommendations in closed populations are to test all residents [16], but how frequent testing is needed is not easily determined.

The suspicion of COVID-19 in hemodialysis patients is not easy. Disease can be oligosymptomatic and PCR repeatedly negative, despite clinical and analytical features very suggestive of COVID-19, and the diagnosis confirmed by antibody detection. Waiting for the seroconversion of PCR- patients may unnecessarily delay the therapy and dynamics of seroconversion in hemodialysis patients is poorly characterized. In this regard, our study provides further data by analyzing IgG and IgM seroconversion in hemodialysis patients. No study has so far analyzed this response in hemodialysis patients. In the general population anti-SARS-CoV antibodies first appear from day 3–42 for IgM and from day 5–47 for IgG antibodies [23], while anti-SARS-CoV-2 antibodies develop from day 9–10 after first symptoms [24]. However, in the prior SARS-CoV epidemic, the mean time for seroconversion was 20 days, at which time only 60–75% of patients had IgG against the virus [25]. In patients with confirmed SARS-CoV-2 infection in China, the mean time for seroconversion was 11–14 days, depending on the immunological assay used and 94.3% (IgM) and 79.8% (IgG) of patients seroconverted after 15 days [26]. In our case, 13% of PCR positive patients had not yet seroconverted at the last follow-up, a mean of 13 days after PCR positivity. As with other infections and vaccines, hemodialysis patients may have a reduced immunogenic response [5].

The most common symptoms of COVID-19 were fever (44% on admission and 89% during hospitalization), fatigue, shortness of breath and dry cough (68%), although many patients also had gastrointestinal symptoms, such as nausea and diarrhea [27]. In our series, PCR positive hemodialysis patients presented fever (31.5%), cough (10.5%) and diarrhea (10.5%) as the most frequent symptoms. Thus, they may be less symptomatic than the general population, although our policy of early testing may have identified milder cases that may have not been diagnosed in a general population setting. In the literature, lymphopenia was present in 83% of COVID-19 patients on admission [28]. Lymphopenia was also a defining characteristic in COVID-19 hemodialysis patients.

Pneumonia is the most frequent complication of COVID-19 [29]. In our series, 56% of hemodialysis patients developed pneumonia as detected by X-ray, the majority of which were bilateral. This is in line with general population Chinese data, in which no X-ray abnormalities were found in 40% of the patients. This is reason why, in China, all patients underwent a chest CT scan, finding pathological abnormalities in 86% of the cases. Three of our hemodialysis patients died of respiratory failure. While a more limited series of five hemodialysis patients with pneumonia described good outcomes, with none of them dying despite all presenting fever, cough and lymphopenia [30], this Chinese report is in line with the findings in our study of a large variability in clinical presentation among hemodialysis patients.

Spontaneous lymphopenia can be found in patients on hemodialysis [31], so it may present interpretation problems for suspected COVID-19. In our series, the mean lymphocyte counts in PCR positive patients in the three months prior to infection was 1300 ± 595 per µL, while median values at diagnosis was 700 per µL. This is in line with prior descriptions of a total lymphocyte decrease in COVID-19, especially if pneumonia develops [32]. Frequently, analytical monitoring in hemodialysis patients means that, contrary to the general population, an acute drop in lymphocytes as compared to the prior control value can be observed. Ferritin was also elevated in PCR positive patients, but did not increase with the severity of the condition. D-dimer also increases in severe COVID-19 patients, indicating vascular damage. However, D-dimer is a marker that is elevated in dialysis patients [13,33].

In populations with preserved renal function, the most frequent cause of death was severe respiratory failure, which causes lung and systemic inflammation, leading to multiorgan failure in patients with comorbidities. Other frequent causes of death were sepsis and heart failure [22,27]. In our study, the mortality pattern presented several aspects that merit discussion, and should be validated in multicenter studies. First, mortality was concentrated in the oldest patients, and in unstable patients who had just started hemodialysis for diseases in which the immune system was additionally compromised by the causative disease itself and/or its treatment. In this regard, it is well known that the mortality rate in patients entering dialysis, especially elderly patients, is higher in the first 90 days of initiating dialysis [34,35]. While it has been discussed to what extent SARS-CoV-2 could be a by-stander in some patients with severe comorbidities, it is also true that these patients are at higher risk of severe COVID-19, and that the worldwide consensus so far has been to attribute the death to COVID-19 if the patient was infected by SARS-CoV-2. Second, mortality was concentrated among PCR positive patients. This would be aligned with the idea that these patients may have higher viral loads. In this regard, the active search of a COVID-19 diagnosis resulted in the identification of individuals with milder disease and, potentially, to an earlier prescription of therapy. This may explain the mortality rate that appeared to be lower than in European centers caring for patients of similar age, and in the range described for Chinese centers caring for patients that were around a decade younger, and in which all hemodialysis patients were screened for COVID-9 (Appendix A).

In COVID-19, there are two distinct but overlapping pathophysiological mechanisms, the first triggered by the virus itself, and the second by the host response. At the time of the cases reported, treatment remained empirical. Drug therapy directed against the virus shows the greatest promise when applied early in the course of the disease, but its usefulness in advanced stages may be questionable [36]. Anti-inflammatory therapy introduced too early may not be necessary, and could even encourage viral replication, as in the case of corticosteroids [37]. In our series, anti-inflammatory therapy was prescribed when there were signs of clinical or analytical severity. Recommended anti-inflammatory therapy includes corticosteroids, IL-6 antagonists (tocilizumab) and JAK inhibitors (Baricitinib) [38].

Several limitations should be acknowledged. Given the small sample size and quasi-single center (two interconnected hemodialysis units) nature of the study, these data may not be generalizable. Additionally, some patients were diagnosed based on serological tests, which have been criticized and two were diagnosed on clinical grounds. The latter is also one of the strengths of the study, since the knowledge that some patients with typical clinical and analytical features can be PCR negative, and may not be tested for antibodies or these may be negative, has strong consequences for preventive and isolation strategies. Indeed, hemodialysis patients are known to mount suboptimal antibody responses to a variety of antigenic challenges. Additionally, there is increasing evidence that, even in the general population, anti-SARS-CoV-2 antibodies may be negative both in the acute and the convalescent phase of the disease, and some patients without detectable antibodies have evidence of infection, as detected by either PCR or specific T-cell reactive cells [39,40]. In this regard, the only two patients to have been diagnosed as having COVID-19, despite negative PCR and serology, had typical COVID-19 features, including severe lymphopenia coincident with symptoms, pneumonia and increased D-Dimer and IL-6 levels. Given the advancing understanding of the disease, it is possible that some patients not considered to have had COVID-19 based on PCR and serology may indeed have had the disease. Finally, as patients did not have a long-term follow-up, the impact of COVID-19 on morbidity and mortality may have been underestimated.

## 5. Conclusions

In conclusion, given uremic immunosuppression and the high prevalence of comorbidities, hemodialysis patients may be particularly vulnerable to COVID-19. The diagnosis is complicated by the presence of PCR-negative patients, and the high incidence of oligosymptomatic individuals. Efforts to identify and treat patients early may limit the spread of the virus and improve outcomes. We suggest that PCR should be performed on hemodialysis patients, including those with mild symptoms, and asymptomatic patients with possible exposure to COVID-19 or affected relatives. The use of this approach resulted in an overall mortality in the range observed for the general population of the same age.

## Figures and Tables

**Figure 1 jcm-09-02264-f001:**
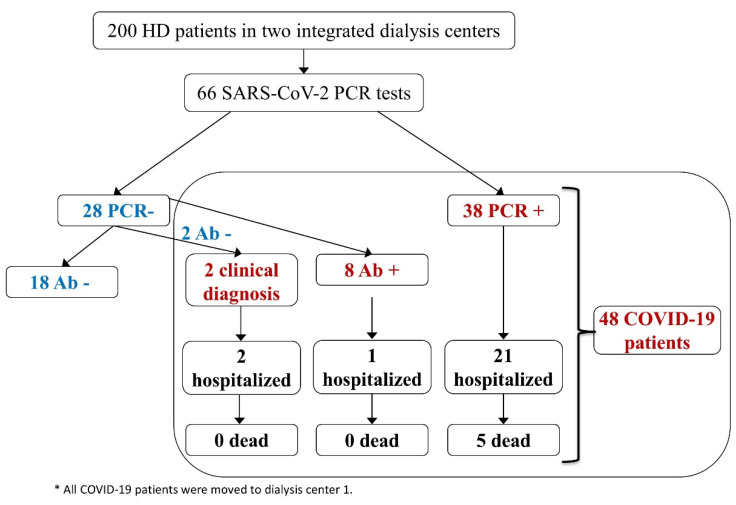
Hemodialisis patients diagnostic flow chart. HD: hemodialysis; PCR +: PCR positive patients; PCR −: PCR negative patients; Ab +: SARS-CoV-2 seropositive patients; Ab −: No antibodies.

**Table 1 jcm-09-02264-t001:** Characteristics of patients with positive and negative polymerase chain reaction (PCR). Data presented as mean ± SD, median (interquartile range) or counts (percentage).

Characteristics	Positive PCR		Negative PCR		*p*-Value
	(*N* = 38)		(*N* = 10)		
Age, years	73.4	±11.86	69.5	±15.3	ns
Time in dialysis, days	1101	(191–1624)	1010	(415–1157)	
Female, *n* (%)	12	(31.6)	4	(40)	ns
Hospital admission, *n* (%)	21	(55.3)	3	(30)	0.042
Symptoms, *n* (%)	33	(86.8)	4	(40)	0.001
Asymptomatic	5	(13.2)	6	(60)	
Oxygen saturation; %	94	(20)	96	(10)	ns
Chest X-ray, *n* (%)					ns
Normal	12	(31.5)	1	(10)	
Unilateral pneumonia	8	(21.1)	1	(10)	
Bilateral pneumonia	18	(44.7)	1	(10)	
No			7		
Laboratory analysis					
Hemoglobin, g/dL	11.28	±1.45	11.40	±1.56	ns
Lymphocytes, /µL ^n^	700	(500–1000)	950	(825–1225)	ns
D dimer ^n^, µg/L ^n^	1127	(710–1772)	1308	(692–1774)	ns
Ferritin, ng/mL	923	(393–1422)	340	(222–807)	0.062
Troponin I ^n^, ng/mL ^n^	0.050	(0.020–0.098)	0.012	(0.001–0.040)	0.033
IL-6 ^n^, pg/mL ^n^	15.40	(7.03–54.25)	15.66	(3.47–21.23)	ns
Serology, *n* (%)	29	(76.3)	27	(96.4)	
IgM, *n* (%)					
Positive	17	(58.6)	8	(25.9)	0.012
Undetermined	4	(13.8)	0	(7.4)	
Negative	8	(27.6)	2	(66.7)	
IgG, *n* (%)					
Positive	17	(58.6)	4	(40)	< .001
Undetermined	4	(13.8)	0	(0)	
Negative	8	(27.6)	6	(60)	
IgG + IgM positive, *n* (%)	15	(51.7)	4	(40)	< .001
Deaths, *n* (%)	5	(13.2)	0		

^n^ Normal range: Lymphocytes 1200–5000/ µL; D-dimer 68–494 µg/L, Troponin I < 0.08 ng/mL, IL6 < 7 pg/mL. PCR: polymerase chain reaction; IL-6: interleukin-6; IgM: immunoglobulin M; IgG: immunoglobulin G; ns: non-significant variables.

**Table 2 jcm-09-02264-t002:** Analysis of polymerase chain reaction (PCR) positive patients, according to hospitalization related to disease severity. Data presented as mean ± SD, median (interquartile range) or counts (percentage).

Characteristics	Hospitalization				*p*-Value
	No (*N* = 17)		Yes (*N* = 21)		
Age, years	72.9	±13.6	73.7	±10.8	ns
Dialysis vintage, days	1011	(1501571)	1360	(251–1614)	
Female, *n* (%)	4	(25)	8	(36)	ns
Symptoms, *n* (%)	12	(68.75)	21	(100)	ns
Asymptomatic	5	(31.25)	0		
Oxygen saturation, %	94.13	±4.78	91.86	±4.84	ns
Chest X-ray, *n* (%)					0.067
Normal	8	(50)	4	(18.2)	
Unilateral pneumonia	4	(25)	4	(18.2)	
Bilateral pneumonia	4	(18.75)	14	(63.6)	
Laboratory analysis					
Hemoglobin, g/dL	11.7	±1.2	11.1	±1.6	0.089
Lymphocytes, /µL ^n^	950	(600–1250)	600	(300–700)	0.007
D dimer ^n,^, µg/L	1045	(499–1379)	1475	(798–2240)	0.17
Ferritin, ng/mL	733	(365–1457)	956	(654–1412)	ns
Troponin I ^n^, ng/mL	0.050	(0.012–0.065)	0.050	(0.025–0.215)	ns
IL-6 ^n^, pg/mL	9.49	(5.74–17.35)	32.20	(11.07–64.92)	ns
Serology, *n* (%)	14	(87.5)	15	(68.2)	
IgM, *n* (%)					ns
Positive	11	(78.6)	8	(53.3)	
Negative	3	(21.4)	7	(46.7)	
IgG, *n* (%)					0.062
Positive	12	(85.7)	7	(46.7)	
Negative	2	(14.3)	8	(53.3)	
IgG + IgM positive, *n* (%)	11	(78.6)	6	(40)	
Deaths, *n* (%)	0		5	(22.7)	

^n^ Normal range: Lymphocytes 1200–5000/ µL; D-dimer 68–494 µg/L, Troponin I < 0.08 ng/mL, IL-6 < 7 pg/mL. IL-6: interleukin-6; IgM: immunoglobulin M; IgG: immunoglobulin G; ns: non-significant variables.

**Table 3 jcm-09-02264-t003:** Analysis of polymerase chain reaction (PCR) negative patients according to hospitalization related to disease severity. Data presented as mean ± SD, median (interquartile range) or counts (percentage).

Characteristics	Hospitalization				*p*-Value
	No (*N* = 7)		Yes (*N* = 3)		
Age, years	66.8	±16.75	76	±10.50	
Time in dialysis, days	1118	(268–3035)	757	(111–1766)	ns
Female, *n* (%)	3	(42.9)	1	(33.3)	ns
Symptoms, *n* (%)	1	(14.3)	3	(100)	
Asymptomatic	6	(85.7)	0		
Oxygen saturation, %	no data		94	±3	
Chest X-ray, *n* (%)					
Normal	1	(14.3)	0		
Unilateral pneumonia	0		1	(33.3)	
Bilateral pneumonia	0		2	(66.7)	
No	6	(85.7)	0		
Clinical analysis					
Hemoglobin, g/dL	11.1	±1.53	12.0	±1.75	
Lymphocytes ^n^, /µL	1000	(900–1350)	400	(300–700)	0.04
D dimer ^n^, µg/L	1512	(577–1754)	1105	(938–1528)	ns
Ferritin, ng/mL	346	(229–730)	335	(217–1432)	ns
Troponin I ^n^, ng/mL	0.026	(0.004–0.040)	0.010	(0.006–0.045)	ns
IL-6 ^n^, pg/mL	3.8	(3.2–15.7)	41.7	(31.2–144.8)	ns
Serology, *n* (%)	7	(100)	3	(100)	
IgM, *n* (%)					ns
Positive	7	(100)	1	(33.3)	
Undetermined	0		0		
Negative	0		2	(66.7)	
IgG, *n* (%)					ns
Positive	3	(42.9)	1	(33.3)	
Undetermined	0		0		
Negative	4	(57.1)	2	(66.7)	
IgG + IgM positive, *n* (%)	3	(42.9)	0		
Deaths, *n* (%)	0		0		

^n^ Normal range: Lymphocytes 1200–5000/ µL; D-dimer 68–494 µg/L, Troponin I < 0.08 ng/mL, IL-6 < 7 pg/mL. IL-6: interleukin-6; IgM: immunoglobulin M; IgG: immunoglobulin G; ns: non-significant variables.

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
