# Peer review of "The Spectrum of Clinical and Serological Features of COVID-19 in Urban Hemodialysis Patients"

_jcm, 2020, doi:10.3390/jcm9072264_

Round 1

Reviewer 1 Report

The article “The Spectrum of Clinical, Microbiological, and Serological Features of COVID-19 in Hemodialysis Patients” by Teresa Stock da Cunha et al., is a retrospective observational study concerning the impact of Covid-19 infection on a cohort of CKD patients undergoing hemodialysis treatment in two Spanish Hospitals.

It shows some interesting observations and a lot of data concerning hemodialysis patients affected by Covid-19. The results are well presented, and the discussion is wide-ranging, but regardless of the good quality of the paper, some points require some further discussion by the authors.

  1. The first question that arises refers to the title where the term "Microbiological" is present, which strictly speaking applies badly to the content of the article given the definition of "microbiological" which concerns to “study of microbes, including bacteria and microscopic fungi and protists including methods to survey, culture, stain, identify, engineer and manipulate microbes”. It would be better not to include “microbiological” in the title.
  2. The authors state that the two centers are interconnected for dialysis treatment, so much so that there is a shift of patients from one to the other according to clinical conditions. Now the two dialysis centers are probably to be considered a single center for care modalities and diagnostic-therapeutic protocols. The question is therefore if the second center may be a satellite limited care center or a center with its own management autonomy, but with patients with a low comorbidity index.
  3. About the routine dialysis treatments carried out in the centers, there are no indications on the modalities (HCO3HD, HDF, on-line HDF, High flux dialysis...), nor whether these have been modified in infected patients. Have patients admitted to ICU maintained the usual treatment or have they been treated with alternative modalities like CRRT, or other?
  4. About Results: the tables presented, especially 1 and 2, contain many uneven data with the risk of becoming dispersed and not easily comparable. It might be advisable to reduce them by showing only the significantly different variables, putting all the remaining information in the supplementary files.
  5. Clinical features: In the description of the main clinical and symptomatological characteristics there are no references to taste and smell changes which in the general population are reported as specific to Covid-19 infection. It is not specified whether these symptoms have not been experienced or investigated in dialysis patients since taste changes are often reported to be quite frequent regardless of Covid-19 infection. In tables 1 and 2 the A-V block is referred to as a symptom, while it should be more properly placed among the objective variables, such as Rx findings.
  6. Outcomes: In the cohort of patients examined, the authors observed low mortality, lower than that reported in several other reports of Covid-19 infection. Besides, 2/5 patients were affected by serious systemic diseases, with a reduced life expectancy on dialysis regardless of Covid-19 infection, which could be only a contributing factor in these cases. The observation period of 2 months probably allowed highlighting only the immediate effects of the virus, excluding medium-long term effects. Some patients with negative PCR and negative antibodies, although symptomatic, cast doubt on whether they were infected with Covid-19. In the paper, there is no report about Vit. D status, as it is well known the effect of Vit.D on the response to infections by modulating the innate and adaptive immune response. In the literature, some reports are suggesting a possible role of Vit.D  plasma levels on mortality rate by Covid-19.
  7. International comparing: Table 4 shows a comparison between the data presented in the article and those found in similar published reports on observations made in Spain, China, and Italy. The comparison is interesting, but the table is designed to make it not immediately readable. It might be more synthetic leaving all the remaining information in the supplementary files.
  8. Discussion: The discussion is wide-ranging and well structured, but some statements need to be clarified. There are some patients in the cohort with poor symptoms. Some of them are either negative PCR or do not present seroconversion. Although in contact with definitely infected patients, and with some mild symptoms, the clinical diagnosis of these patients may eventually be reconsidered? The policy of the center to consider these patients as infected too has undoubtedly proved successful in reducing infection and perhaps mortality, but it puts some doubts on the epidemiological data.                                    The authors state that the deaths in cohort patients mainly affected older patients with more severe comorbidities as described in the literature, but also patients who had recently started dialysis treatment. On the other hand, it is well known that the mortality rate in patients entering dialysis, especially elderly patients, is higher in the first 90 days of treatment. In these cases was the role of Covid-19 as the main actor or just an innocent bystander?
  9. A Language revision is necessary owing to several printing errors.

Author Response

The article “The Spectrum of Clinical, Microbiological, and Serological Features of COVID-19 in Hemodialysis Patients” by Teresa Stock da Cunha et al., is a retrospective observational study concerning the impact of Covid-19 infection on a cohort of CKD patients undergoing hemodialysis treatment in two Spanish Hospitals.

It shows some interesting observations and a lot of data concerning hemodialysis patients affected by Covid-19. The results are well presented, and the discussion is wide-ranging, but regardless of the good quality of the paper, some points require some further discussion by the authors.

  1. The first question that arises refers to the title where the term "Microbiological" is present, which strictly speaking applies badly to the content of the article given the definition of "microbiological" which concerns to “study of microbes, including bacteria and microscopic fungi and protists including methods to survey, culture, stain, identify, engineer and manipulate microbes”. It would be better not to include “microbiological” in the title.
  1. We thank the reviewer for this suggestion, that we have followed. Likely the confusion is due to the fact the viral studies are performed by the Microbiology department at our institution.

  1. The authors state that the two centers are interconnected for dialysis treatment, so much so that there is a shift of patients from one to the other according to clinical conditions. Now the two dialysis centers are probably to be considered a single center for care modalities and diagnostic-therapeutic protocols. The question is therefore if the second center may be a satellite limited care center or a center with its own management autonomy, but with patients with a low comorbidity index.

R: This is now more clearly explained in the text. One center is a hospital-based hemodialysis unit that, as the reviewer suggests, generally, but not always, cares for patients with more comorbidity. The second center is a satellite limited care center (providing only hemodialysis outside a hospital setting) with its own management autonomy, that generally, but not always, cares for patients with a lower comorbidity index.

  1. About the routine dialysis treatments carried out in the centers, there are no indications on the modalities (HCO3HD, HDF, on-line HDF, High flux dialysis...), nor whether these have been modified in infected patients. Have patients admitted to ICU maintained the usual treatment or have they been treated with alternative modalities like CRRT, or other?

R: Routine dialysis treatment is provided by either on-line hemodiafiltration or high flux hemodialysis. Routine dialysis treatment was not modified in infected patients, except for infection control measures consisting in distancing and personal protection equipment (PPE). The single patient admitted to the ICU was dialyzed using continuous hemodiafiltration to reduce hemodynamic instability. This information has now been incorporated into the manuscript.

  1. About Results: the tables presented, especially 1 and 2, contain many uneven data with the risk of becoming dispersed and not easily comparable. It might be advisable to reduce them by showing only the significantly different variables, putting all the remaining information in the supplementary files.
  1. As suggested, tables have now been simplified by removing information of dialysis center and symptoms, which have been moved to supplementary data: new tables S2, S3 and S4

  1. Clinical features: In the description of the main clinical and symptomatological characteristics there are no references to taste and smell changes which in the general population are reported as specific to Covid-19 infection. It is not specified whether these symptoms have not been experienced or investigated in dialysis patients since taste changes are often reported to be quite frequent regardless of Covid-19 infection. In tables 1 and 2 the A-V block is referred to as a symptom, while it should be more properly placed among the objective variables, such as Rx findings.
  1. The referee is correct to pointing out the high frequency of taste and smell changes. However, this was not apparent very early in the course of the pandemic and these data were not collected. We agree that the A-V block is a cause for symptoms rather than a symptom in itself and has now been moved to a different section.

  1. Outcomes: In the cohort of patients examined, the authors observed low mortality, lower than that reported in several other reports of Covid-19 infection. Besides, 2/5 patients were affected by serious systemic diseases, with a reduced life expectancy on dialysis regardless of Covid-19 infection, which could be only a contributing factor in these cases. The observation period of 2 months probably allowed highlighting only the immediate effects of the virus, excluding medium-long term effects.

R: the referee is correct that the mid- and longer term consequences of COVID-19 are not well characterized. The lack of follow-up is now further acknowledged as a limitation.

  1. Some patients with negative PCR and negative antibodies, although symptomatic, cast doubt on whether they were infected with Covid-19.
  1. The referee raises this issue also in comment number 10, where an extensive response is provided, indicating how the text was changed.

  1. In the paper, there is no report about Vit. D status, as it is well known the effect of Vit.D on the response to infections by modulating the innate and adaptive immune response. In the literature, some reports are suggesting a possible role of Vit.D  plasma levels on mortality rate by Covid-19.

R: Indeed, the role of vitamin D seems to us very relevant and was requested in most of the patients in the study. But it has not been included because, on the one hand, we do not have analytical data from all the patients and, on the other hand, the data we have will be combined in a multicenter trial to assess the role of calcium metabolism and vitamin D in COVID-19. 

  1. International comparing: Table 4 shows a comparison between the data presented in the article and those found in similar published reports on observations made in Spain, China, and Italy. The comparison is interesting, but the table is designed to make it not immediately readable. It might be more synthetic leaving all the remaining information in the supplementary files.

R: As suggested by the reviewer, the table has been reformatted and is now presented as suppl files

  1. Discussion: The discussion is wide-ranging and well structured, but some statements need to be clarified. There are some patients in the cohort with poor symptoms. Some of them are either negative PCR or do not present seroconversion. Although in contact with definitely infected patients, and with some mild symptoms, the clinical diagnosis of these patients may eventually be reconsidered? The policy of the center to consider these patients as infected too has undoubtedly proved successful in reducing infection and perhaps mortality, but it puts some doubts on the epidemiological data.
  1. We concur that patients with negative PCR and negative antibodies may cast doubt on whether they actually had COVID-19. However, PCR has a sensitivity of only 70% and both PCR and antibody response have time-windows in which they may be negative. Additionally, there is increasing evidence that even in the general population, anti-SARS-CoV-2 antibodies may be negative both in the acute and the convalescent phase of the disease and some patients without detectable antibodies have evidence of infection as detected by either PCR or specific T-cell reactive cells (10.1038/s41591-020-0965-6; https://www.biorxiv.org/content/10.1101/2020.06.29.174888v1). In this regard, the only two patients to have been diagnosed as COVID-19 despite negative PCR and serology had typical COVID-19 features, including severe lymphopenia coincident with symptoms, pneumonia and increased D-Dimer and IL-6 levels (Table S1). Our concern is that some patients not considered to have had COVID-19 based on PCR and serology may indeed have had the disease. This line of reasoning and new references have been now incorporated into the manuscript.

  1. The authors state that the deaths in cohort patients mainly affected older patients with more severe comorbidities as described in the literature, but also patients who had recently started dialysis treatment. On the other hand, it is well known that the mortality rate in patients entering dialysis, especially elderly patients, is higher in the first 90 days of treatment. In these cases was the role of Covid-19 as the main actor or just an innocent bystander?

R: We thank the reviewer for pointing this out. It fits into the wider discussion as to what extent a variable percentage of patients dying from COVID-19 had prior comorbidities and COVID-19 may only have accelerated the inevitable. This is now discussed as follows:” In this regard, it is well known that the mortality rate in patients entering dialysis, especially elderly patients, is higher in the first 90 days of initiating dialysis (Ref). While it has been discussed to what extent SARS-CoV-2 could be a by-stander in some patients with severe comorbidities, it is also true that these patients are at higher risk of severe COVID-19 and that the worldwide consensus so far has been to attribute the death to COVID-19 if the patient was infected by SARS-CoV-2.”

  1. A Language revision is necessary owing to several printing errors.

R: We thank the reviewer for pointing this out. A thorough language revision has been performed.

Reviewer 2 Report

Congratulations dear colleges, dedication and precision does make a difference!

A great work of encouragement of what can be achieved by vigorous efforts. The issue covered of cause is of utmost importance.

The paper gives an overview of a group of dialysis patients in Madrid.

The Title “The Spectrum of Clinical, Microbiological, and Serological Features of COVID-19 in Hemodialysis Patients” seems a little over the top when describing a circumscripted setting in a corona hotspot. Maybe you can find a title more precise.

In my opinion the overall readability and clarity of the text could be somewhat smoothed, focusing on the abstract and the introduction.

E.g.:

Of 200 patients on hemodialysis, 48 (24%) were diagnosed of COVID-19: 38 (79%) 22 were PCR positive, 8 (17%) PCR negative and seroconverted and 2 (4%) were diagnosed on 23 clinical grounds.

Sounds a little odd, how about something like:

In 200 haemodialysis patients COVID-19 was diagnosed in 48, of whom 22 were PCR positive, 8 PCR negative but seroconverted and 2 were diagnosed on typical clinical grounds.

There are few typos and in my copy the tables are shifted and distorted.

The discussion is of great concern, especially the considerations on the patients with the highest levels of risk. But what I sorely miss is a technical proposal or a detailed description on how you acted on the problem of missing signs of infection and negative PCR in your patients.

We used exactly the same approach for early diagnosis of COVID. And we, as you probably did, strictly separated the single shifts of dialysis with no overlap, gaining the time for a thorough disinfection of rooms and equipment.

We separated patients in positive, questionable and probably healthy and strictly assigned the stuff according to their individual risk. We applied constant retraining courses on basic and advanced hygiene concepts and ended up with zero spread of disease in our unit, but of cause faced a less hostile environment with a smaller amount of infection in the general population.

Author Response

  1. The paper gives an overview of a group of dialysis patients in Madrid. The Title “The Spectrum of Clinical, Microbiological, and Serological Features of COVID-19 in Hemodialysis Patients” seems a little over the top when describing a circumscripted setting in a corona hotspot. Maybe you can find a title more precise.

R: We are struggling with this. We may pinpoint the geographical location to Spain, given that many of the readers may not be familiar with the location of the city of Madrid. However, other Spanish cities were not represented. In this regard, a search for the literature suggested that it would be better suited to pinpoint the nature of the dialysis center as “Urban”, as in “Corbett RW, Blakey S, Nitsch D, et al. Epidemiology of COVID-19 in an Urban Dialysis Center [published online ahead of print, 2020 Jun 19]. J Am Soc Nephrol. 2020;ASN.2020040534. doi:10.1681/ASN.2020040534” published in the top ranking nephrology journal

  1. In my opinion the overall readability and clarity of the text could be somewhat smoothed, focusing on the abstract and the introduction. E.g.: Of 200 patients on hemodialysis, 48 (24%) were diagnosed of COVID-19: 38 (79%) 22 were PCR positive, 8 (17%) PCR negative and seroconverted and 2 (4%) were diagnosed on 23 clinical grounds. Sounds a little odd, how about something like: In 200 haemodialysis patients COVID-19 was diagnosed in 48, of whom 22 were PCR positive, 8 PCR negative but seroconverted and 2 were diagnosed on typical clinical grounds.

R: We thank the reviewer for the suggestion that we have adopted.

  1. There are few typos and in my copy the tables are shifted and distorted.

R: We thank the reviewer for pointing this out. A thorough language revision has been performed. Additionally, table 4 has now been moved to suppl data (Table S5) in order to present it horizontally and avoid distortion

  1. The discussion is of great concern, especially the considerations on the patients with the highest levels of risk. But what I sorely miss is a technical proposal or a detailed description on how you acted on the problem of missing signs of infection and negative PCR in your patients. We used exactly the same approach for early diagnosis of COVID. And we, as you probably did, strictly separated the single shifts of dialysis with no overlap, gaining the time for a thorough disinfection of rooms and equipment. We separated patients in positive, questionable and probably healthy and strictly assigned the stuff according to their individual risk. We applied constant retraining courses on basic and advanced hygiene concepts and ended up with zero spread of disease in our unit, but of cause faced a less hostile environment with a smaller amount of infection in the general population.

R: Patients positive for PCR or serology were dialyzed in a room set up for this purpose, completely independent (both in terms of staff and equipment) from non-Covid patients